# Zearalenone Exposure Disrupts STAT-ISG15 in Rat Colon: A Potential Linkage between Zearalenone and Inflammatory Bowel Disease

**DOI:** 10.3390/toxins15060392

**Published:** 2023-06-09

**Authors:** Haonan Ruan, Jiashuo Wu, Fangqing Zhang, Ziyue Jin, Jiao Tian, Jing Xia, Jiaoyang Luo, Meihua Yang

**Affiliations:** 1Key Laboratory of Bioactive Substances and Resources Utilization of Chinese Herbal Medicine, Ministry of Education, Institute of Medicinal Plant Development Chinese Academy of Medical Sciences & Peking Union Medical College, Beijing 100193, China; 2Peking-Tsinghua Center for Life Sciences, Academy for Advanced Interdisciplinary Studies, Peking University, Beijing 100091, China; 3School of Basic Medical Science, Peking University, Beijing 100191, China

**Keywords:** zearalenone, proteomics, colon toxicity, STAT, inflammatory bowel disease

## Abstract

Zearalenone (ZEN), a prevalent mycotoxin contaminating food and known for its intestinal toxicity, has been suggested as a potential risk factor for inflammatory bowel disease (IBD), although the exact relationship between ZEN exposure and IBD remains unclear. In this study, we established a rat model of colon toxicity induced by ZEN exposure to investigate the key targets of ZEN-induced colon toxicity and explore the underlying connection between ZEN exposure and IBD. Histological staining of the rat colon revealed significant pathological changes resulting from ZEN exposure (*p* < 0.01). Furthermore, the proteomic analysis demonstrated a notable upregulation of protein expression levels, specifically STAT2 (0.12 ± 0.0186), STAT6 (0.36 ± 0.0475) and ISG15 (0.43 ± 0.0226) in the rat colon (*p* < 0.05). Utilizing bioinformatics analysis, we combined ZEN exposure and IBD clinical sample databases to reveal that ZEN exposure may increase the risk of IBD through activation of the STAT-ISG15 pathway. This study identified novel targets for ZEN-induced intestinal toxicity, providing the basis for further study of ZEN exposure to IBD.

## 1. Introduction

ZEN, an estrogen-like mycotoxin produced by Fusarium fungi, poses a serious threat to human health and economic development worldwide through its contamination of food and feed [1]. A report from China in 2021 revealed that out of 2626 food samples tested, the positive rate of ZEN was 79.09%, with an average contamination level of 197.37 μg/kg and a maximum contamination level of 10,467 μg/kg [2]. Iwase et al. found that 60% of Brazilian barley samples were contaminated with *Fusarium*, with ZEN contamination levels ranging from 74 to 556 µg/kg, surpassing the maximum residue limit set in Brazil and Europe in 30% of the samples [3]. Importantly, ZEN exposure has been associated with various toxic effects in humans, including reproductive toxicity, intestinal toxicity and hepatotoxicity [4]. Current research focuses on studying the toxicity and metabolism of ZEN and its metabolites [5,6]. ZEN is classified as a Group III carcinogen by the World Health Organization (WHO), prompting countries and organizations worldwide to establish limit standards for ZEN in food and feed [7].

The gut, as the first organ exposed to ZEN, is particularly susceptible to its toxic effects [8]. Research has demonstrated that ZEN-induced intestinal toxicity involves multiple mechanisms, with inflammation playing a critical role [9]. ZEN triggers intestinal inflammation by interfering with TLR4, an upstream target of NF-κB [10,11]. Additionally, ZEN activates reactive oxygen species-mediated NLRP3 inflammasomes in IPEC-J2 cells, leading to the activation of caspase-1-dependent inflammatory factors interleukin (IL)-1β and IL-18 [12]. A study by Wang et al. showed that ZEN promotes intestinal inflammation by disrupting the reproductive and immune axis and altering the levels of intestinal microbial metabolites in pigs [13].

Inflammatory bowel disease, comprising Crohn’s disease and ulcerative colitis, is a growing global healthcare challenge with increasing incidence rates [14]. Despite extensive research, the exact precise of IBD remains largely elusive. However, it is widely believed that the disease arises from a complex interplay of genetic, environmental and microbial factors, leading to immune dysregulation [15]. Recent studies have revealed a close association between mycotoxin exposure (including ZEN) and the development and progression of IBD [16,17,18,19]. For instance, the fusarium mycotoxin deoxynivalenol (DON) has been shown to disrupt the balance of T helper/Treg cells in the intestine, increasing the risk of IBD [20]. Additionally, fecal macrogenomics and metabolomics studies have demonstrated that AFB_1_ disrupts intestinal microbiota homeostasis and may play a significant role in the pathogenesis of IBD [21].

While it is established that ZEN can induce severe intestinal inflammation, the connection between ZEN exposure and IBD remains unclear. To further investigate the induction of colonic toxicity by ZEN and identify the targets involved, we developed a rat model of ZEN-induced colonic toxicity and performed proteomic analysis to elucidate the critical factors. Moreover, by integrating the genes affected by ZEN exposure with genomic data from clinical samples of IBD, we conducted a bioinformatics analysis to uncover potential associations between ZEN exposure and IBD. This study provides the basis for further study of ZEN exposure to IBD.

## 2. Results

### 2.1. ZEN Exposure Induces Histopathological Changes in the Colon

To investigate the impact of ZEN on the colon of rats, we assessed colon length and performed hematoxylin and eosin (HE) staining to evaluate colon tissue in different groups. The results of our experiment demonstrated that rats exposed to 5 mg/kg/d b.w. ZEN (9.84 ± 0.7820, 8.8–11.1 cm) exhibited a significant reduction in colon length compared to the control group (6.99 ± 1.0785, 5.1–8.7 cm) (Figure 1A). HE staining revealed that the colon tissue of the control group rats exhibited a complete and clear structure, with well-organized and dense villi, regular gland arrangement and no mucosal necrosis or shedding. Conversely, ZEN exposure led to colon lumen enlargement, crypt atrophy, villus breakage, loss of goblet cells, infiltration of inflammatory cells, mucosal swelling, disrupted tissue structure and glandular swelling (Figure 1B).

Based on the HE staining results, we conducted further analysis of villus height, crypt depth, epithelial thickness and histological damage score in colon slices from each group of rats. Our findings demonstrated that the ZEN group exhibited significantly lower villus height compared to the control group (*p* < 0.01) (Figure 1C), while the crypt depth was significantly higher in the ZEN group compared to the control group (*p* < 0.05) (Figure 1D). Additionally, ZEN exposure led to a significant decrease in the villus height/crypt depth ratio (*p* < 0.01) (Figure 1E). Moreover, the epithelial thickness of the colon in rats was significantly reduced following ZEN exposure compared to the control group (*p* < 0.01) (Figure 1F). In summary, the histological damage score of the rats’ colons indicated that ZEN exposure caused severe colon damage compared to the control group (*p* < 0.01) (Figure 1G).

### 2.2. ZEN Exposure Alters the Proteomics of the Colon

To investigate the impact of ZEN exposure on protein levels in the rat colon, we conducted proteomic experiments on colon tissues from rats in each group. Subsequently, we analyzed and compared the proteomic profiles between the ZEN and control groups. OPLS-DA analysis was employed to distinguish the proteomes of the two groups, yielding favorable model parameters (R^2^X = 0.914, R^2^Y = 0.999, Q^2^ = 0.506) (Figure 2A–C). By combining log_2_FC and *p*-value criteria (FC > 1.2 or FC < 0.83, *p*-value < 0.05), we identified 35 differentially expressed proteins (DEPs) in the ZEN group, consisting of 9 downregulated and 26 upregulated proteins (Figure 2D). We further performed cluster heatmap analysis using these DEPs (Table 1) (Figure 2E). The proteomic analysis comparing ZEN and control groups revealed significant alterations in protein expression levels in the rat colon induced by ZEN exposure.

### 2.3. ZEN Exposure Affects Proteasome Mediated Ubiquitination in the Colon

To gain further insight into the effect of ZEN exposure on protein function in the rat colon, we conducted KOG and Gene Ontology (GO) enrichment analyses on the DEPs between the ZEN and control groups. The subcellular localization analysis revealed that 51.43% of DEPs were located in the cytoplasm, while both nuclear and extracellular DEPs accounted for 14.29% (Figure 3A). KOG enrichment analysis highlighted that the DEPs primarily functioned in posttranslational modification, protein turnover and chaperones (Figure 3B).

In the biological process enrichment analysis of GO analysis, we observed that the DEPs were primarily involved in catalytic processes, including protein catabolic processes, organonitrogen compound catabolic processes, and macromolecule catabolic processes. Notably, the catalytic processes encompassed ubiquitin-dependent catalytic processes, protesome-mediated ubiquitin-dependent catalytic processes, and proteasome-mediated ubiquitin-dependent catalytic processes (Figure 3C). The cell component enrichment analysis of GO analysis indicated that the DEPs were predominantly associated with catalytic complexes, proteasome complexes, proteasome core complexes and proteasome accessory complexes (Figure 3D). Furthermore, the molecular function enrichment analysis of GO analysis suggested that the DEPs were mainly involved in catalytic activity, acting on proteins, ubiquitin-protein transfer activity and ubiquitin-like protein transfer activity (Figure 3E).

Based on the results of the enrichment analysis, we identified that the DEPs in the ZEN group were closely related to the proteasome-mediated ubiquitination pathway. Notably, after screening, we discovered two key proteins associated with proteasome-mediated ubiquitination: ISG15 (Figure 3F) and PSMC4 (Figure 3G) [22,23]. In comparison to the control group, ZEN exposure significantly increased the protein expression levels of ISG15 (*p* < 0.05) and PSMC4 (*p* < 0.05) in the rat colon. These findings indicate that ZEN exposure activates proteasome-mediated ubiquitination in the rat colon.

### 2.4. ZEN Exposure Affects the Expression Levels of STAT Proteins in the Colon

To further investigate the key targets of ZEN-induced colon toxicity in rats, we performed domain and Kyoto Encyclopedia of Genes and Genomes (KEGG) enrichment analyses, as well as protein-protein interaction (PPI) analysis, on the DEPs between the ZEN and control groups. The structural domain enrichment analysis revealed that the DEPs were primarily associated with STAT and proteasome structural domains (Figure 4A). KEGG enrichment analysis of the DEPs demonstrated that ZEN exposure significantly activated the JAK-STAT signaling pathway and the proteasome pathway in the rat colon (Figure 4B,C). Consistent with these findings, the PPI analysis showed significant interactions among the proteins STAT2, STAT6, ISG15 and PSMB8 within the DEPs (Figure 4D). In comparison to the control group, ZEN exposure led to a significant upregulation in the protein expression of STAT6 (Figure 4E) (*p* < 0.05) and STAT2 (Figure 4E) (*p* < 0.05). These results suggest that ZEN exposure induces intestinal inflammation by activating the JAK-STAT pathway, which may contribute to the disruption of the proteasome-mediated ubiquitination pathway in the rat colon.

### 2.5. ZEN Exposure May Increase the Risk of IBD by Disrupting STAT Family

Our experimental findings demonstrate that ZEN exposure affects the STAT family and the proteasome-mediated ubiquitination pathway in the rat colon. To further investigate and predict the potential association between ZEN exposure and the development of IBD, we utilized the gene expression omnibus (GEO) platform to screen relevant databases containing ZEN exposure and IBD clinical samples. These datasets were normalized, filtered and intersected to identify the cross-differentially expressed genes (cDEGs) associated with both ZEN exposure and IBD (Figure 5A). Subsequently, the cDEGs were subjected to GO and KEGG enrichment analyses, as well as PPI analysis, using the Metascape and STRING platforms, respectively.

The biological process of GO enrichment analysis of the cDEGs (Figure 5B) showed that ZEN exposure may increase the risk of IBD through response processes (response to chemicals, response to oxygen-containing compound, response to hormones and response to lipid, etc.), positive regulation of protein catabolic process and immune regulation processes (regulation of leukocyte cell−cell adhesion, negative regulation of α-β T cell activation, negative regulation of CD4-positive, α-β T cell and activation of NF-κB−inducing kinase activity, etc.) KEGG pathway enrichment analysis of cDEGs (Figure 5C) showed that ZEN exposure might increase the risk of IBD through pathways in cancer, MAPK signaling pathway, JAK−STAT signaling pathway and IL-17 signaling pathway, etc. GO and KEGG enrichment analysis of the cDEGs indicated that ZEN exposure might increase the risk of IBD by disrupting the JAK-STAT-mediated inflammation and immune response. PPI analysis of the cDEGs (Figure 5D) showed that the top 5 genes with the highest scores included SRC, STAT1, AR, CD44 and RUNX1. Therefore, we speculate that ZEN exposure may increase the risk of IBD by affecting inflammation by disrupting the STAT family.

## 3. Discussion

Mycotoxins are fungal metabolites that can contaminate food and the environment [24]. Human exposure to mycotoxins occurs through the consumption of contaminated food, inhalation of mycotoxin-contaminated air and contact with mycotoxin-contaminated soil and water sources [25]. IBD comprises chronic and complex conditions such as ulcerative colitis and Crohn’s disease [26]. Recent research suggests a potential link between mycotoxins and the onset and progression of IBD, with mycotoxin exposure exacerbating the symptoms in patients with CD and UC [27]. Furthermore, the fungal composition in individuals with IBD differs from that in healthy individuals, and some of these strains may produce mycotoxins [28]. Patients with IBD may be more susceptible to mycotoxin exposure due to the inflammatory state of their intestines, which can increase intestinal permeability and facilitate the entry of mycotoxins into the bloodstream, triggering immune responses and intestinal inflammation [29].

ZEN, commonly found in various grains and foods, particularly corn, have been associated with various health issues, including cancer, immune system suppression and digestive problems [30,31]. Recent studies have identified a potential correlation between ZEN exposure and IBD. ZEN has been found to induce intestinal epithelial cell death and exacerbate inflammation in IBD [32]. Moreover, ZEN exposure can disrupt the balance of gut microbes, leading to dysbiosis and intestinal inflammation, thereby increasing the risk of IBD [33]. Additionally, ZEN exposure can impact the occurrence and progression of IBD by activating genes or proteins associated with the disease [34]. Through a combination of colon proteomics and analysis of clinical samples, our experiment revealed that ZEN exposure might elevate the risk of IBD by upregulating the expression levels of STAT2, STAT6 and ISG15 proteins.

Recent studies have highlighted the significant roles of STAT and ISG15 proteins in the development of IBD [35,36]. Increased expression levels of STAT have been observed in the intestinal tissues of IBD patients [37]. STAT proteins can activate transcription factors such as NF-κB and AP-1, promoting the expression of various inflammation-related genes, including IL-6, TNF-α and IL-12 [38]. Abnormal expression and production of these inflammatory factors are key pathological features of IBD [39]. Hence, elevated STAT expression may contribute to the occurrence and progression of IBD [40]. ISG15 also plays a role in IBD, as its expression in the intestinal tissues of IBD patients is increased and positively correlated with disease severity [41]. ISG15 regulates immune cell proliferation, differentiation and apoptosis, thereby participating in the modulation of intestinal immune responses [42]. Additionally, ISG15 can influence cell apoptosis and the activation of signaling pathways such as NF-κB, further impacting inflammatory reactions and cell proliferation [43].

Notably, there exists an interaction between STAT and ISG15 [44]. Studies have revealed that ISG15 expression can influence the activation of the STAT1 signaling pathway by modulating the phosphorylation and activation state of STAT1 [45]. Similarly, STAT3 can regulate the expression levels of ISG15, thereby influencing the function of ISG15 [46]. In active ulcerative colitis and Crohn’s disease, intestinal epithelial cells express immunomodulatory ISG15 via JAK1-STAT-IRF9 [47]. These reciprocal regulatory relationships may exert an impact on the occurrence and development of IBD. Both STAT and ISG15 can be activated by inflammatory cytokines such as interleukins and interferons. Previous studies have demonstrated that exposure to ZEN enhances the production of intestinal inflammatory cytokines [48]. Additionally, a non-toxic dose of DON has been found to exacerbate DSS-induced colitis through the JAK2/STAT3 signaling pathway, suggesting its association with IBD risk [49]. Our experiments revealed that ZEN exposure induced the expression of STAT and ISG15 in rat colon, indicating that ZEN exposure may elevate the risk of IBD by activating the IFN-STAT-ISG15 pathway (Figure 6). However, this study only reported the results of ZEN exposure leading to colon toxicity in rats, and further experimental research is needed to establish evidence between ZEN and human IBD.

## 4. Conclusions

Numerous studies have indicated that ZEN, a prevalent mycotoxin found in food, may pose a risk for IBD, although its exact mechanism remains unclear. In this study, we established a rat model of ZEN-induced colonic toxicity to investigate the key targets associated with ZEN exposure and colonic toxicity using proteomics. Furthermore, we inferred the potential link between ZEN exposure and IBD by integrating bioinformatics analysis with clinical sample data from IBD patients. Our findings revealed that ZEN exposure significantly increased the expression levels of STAT2, STAT6 and ISG15 proteins in the rat colon. Based on the bioinformatics analysis, we propose that ZEN exposure may elevate the risk of IBD by activating the IFN-STAT-ISG15 axis.

## 5. Materials and Methods

### 5.1. Chemicals and Reagents

ZEN was obtained from Yuanye (Shanghai, China). Ethanol, acetic acid and hydrochloric acid were obtained from Sigma-Aldrich (Shanghai, China).

### 5.2. Animal Study

The animal experiment followed the guidelines outlined in the Guide for the Care and Use of Laboratory Animals published by the US National Institutes of Health and the ARRIVE guidelines. The animal experiment was approved by the Laboratory Animal Center at the Institute of Medicinal Plant Development, Chinese Academy of Medical Sciences and Peking Union Medical College (Approval No. SLXD-20210422034, 22 April 2021, Beijing, China). Twenty 8-week-old female SD rats (200 ± 20 g) were obtained from Beijing Vital River Laboratory Animal Technology Co., Ltd. They were kept in a room with a 12-h light/dark cycle and maintained at a constant temperature (23 ± 3 °C) and humidity (50 ± 10%). The rats were housed in a room with a 12-h light/dark cycle and maintained at a constant temperature (23 ± 3 °C) and humidity (50 ± 10%). They were provided with UV-disinfected fodder (NCD, Beijing Hfk Bioscience Co., Ltd., Beijing, China) and water ad libitum. A one-week adaptive breeding period was conducted prior to the experiments. We strictly followed the exclusion criteria specified in the guidelines, and no animals, experimental units or data points were excluded from the experiment.

The rats were randomly divided into two groups: the control group (*n* = 10) and the ZEN group (*n* = 10). The ZEN group received gavage administration of ZEN (5 mg/kg/d b.w., dissolved in 0.3% sodium carboxymethyl cellulose), while the control group received an equivalent volume of vehicle (0.3% sodium carboxymethyl cellulose). This administration was carried out for a period of 14 days [50]. Throughout the experiment, the animals had unrestricted access to food and drinking water. Daily monitoring included recording body weight and food intake. After 16 h of fasting, the rats were intraperitoneally injected with urethane (20%, 1 g/kg) for anesthesia. Tissues were weighed, and colons were carefully harvested and measured (the measured colon segment is located below the cecum and above the rectum) [51]. Subsequently, the samples were stored at −80 °C until analysis.

### 5.3. HE Staining

HE staining was performed as described previously [52]. Fresh tissue was fixed with 4% paraformaldehyde for 48 h and then trimmed and placed in an embedding box. The tissue was subjected to a gradient alcohol dehydration process with 70%, 80%, 95% and 100% alcohol for 30 min each, followed by two rounds of xylene treatment for 20 min each and two rounds of paraffin-soaked wax for 12 min each. Paraffin embedding was carried out using a paraffin embedding machine. After embedding, the tissue blocks were sectioned using a microtome to obtain approximately 4 μm thick sections. The paraffin-embedded sections were subsequently stained with HE. To deparaffinize the sections, three changes of xylene were used for 8 min each, followed by two changes of 100% alcohol for 8 min each, and 90%, 80% and 60% alcohol for 8 min each. The sections were then stained with Hematoxylin for 4 min and rinsed with running water. Differentiation was performed using hydrochloric acid alcohol for 2–3 s, followed by water rinsing and a 20-s treatment with 0.5% ammonia water for observation. Eosin staining was applied for 1 min. Differentiation was performed using 80% and 90% alcohol for 3–5 s, 95% alcohol for 5 min, three changes of 100% alcohol for 5 min each, and two changes of xylene for 5 min each. Finally, the neutral resin was used for sealing, and histopathological changes were observed using a microscope imaging system. Classical visual field images that showed complete villi in a straight direction were selected for data collection. The height of the intestinal villi and the depth of the crypt were measured for statistical analysis. Histologic scoring was performed based on the criteria presented in Table 2 [53].

### 5.4. Proteomic Analysis of Rat Colon

Proteomic analysis of rat ovaries was performed and analyzed as previously described [54] following standard operating procedures. Colon samples from rats underwent pretreatment, and the protein concentration was determined using a BCA kit. The pretreated samples were then digested and desalted using a C18 solid-phase extraction column. Peptides were separated using the EASY-nLC 1200 ultra-performance liquid chromatography system (ThermoFisher Scientific, Waltham, MA, USA) and analyzed using nano-electrospray ionization on a Q ExactiveTM HF-X mass spectrometer (ThermoFisher Scientific). The resulting MS/MS data were analyzed using the MaxQuant search engine (version 1.6.15.0). Bioinformatics analysis of the proteomic data was performed as previously described [54].

### 5.5. Gene Expression Data Set

We obtained specific gene expression raw data sets related to ZEN exposure and IBD from the GEO at the National Center for Biotechnology Information. The GEO accession numbers for the ZEN exposure data sets are GSE5734, GSE14774 and GSE119552. Furthermore, we included the genes identified as DEPs from the proteomics analysis conducted in this experiment in the ZEN exposure data sets. The GEO accession numbers for the IBD data sets are GSE1141, GSE1142, GSE1152, GSE6731, GSE36807, GSE59071 and GSE72780. The samples in these data sets were collected from gut samples obtained from individuals with IBD as well as healthy control subjects.

### 5.6. Identification of DEGs

The obtained data were processed using various R packages (version: 3.6.2) within RStudio. Each dataset underwent quality control measures to minimize the false detection rate following the procedures outlined in the original study. The expression matrix was normalized using the normalizeBetweenArrays function from the limma package (version: 3.42.2) [55]. The normalized data were then subjected to analysis using the limma package to identify genes that showed differential expression between samples exposed to ZEN and those treated with the control vehicle, as well as between IBD samples and their corresponding control samples. Genes with a *p*-value < 0.05 and a log_2_ fold change (FC) of ≥1.2 were considered as DEGs in accordance with the criteria specified in the original study.

### 5.7. Gene Set Enrichment Analysis and PPI Analysis

The cDEGs were subjected to further functional enrichment analysis. To perform this analysis, the cDEGs were uploaded to the Metascape database (https://www.metascape.org/, accessed on 13 April 2022). The Metascape database provided predictions and mappings of GO enrichment and KEGG pathways associated with the cDEGs [56]. To establish a PPI network for the cDEGs, the STRING database (version 11.5) (https://www.string-db.org/ accessed on 13 April 2022) was utilized with default conditions. The resulting interaction file, which contained the source and target nodes, was then imported into Cytoscape software (version 3.7.1). In Cytoscape, the MCODE plug-in unit [57] was used to identify gene clusters within the PPI network.

### 5.8. Statistical Analyses

T Statistical analyses were performed using GraphPad Prism 8.0.2 software. Replicate measurements were obtained from separate biological samples, and the data are presented as mean ± standard deviation. The statistical significance was assessed using a student’s *t*-test, with a probability value (*p*-value) of less than 0.05 considered statistically significant and a *p*-value of less than 0.01 considered statistically highly significant.

## Figures and Tables

**Figure 1 toxins-15-00392-f001:**
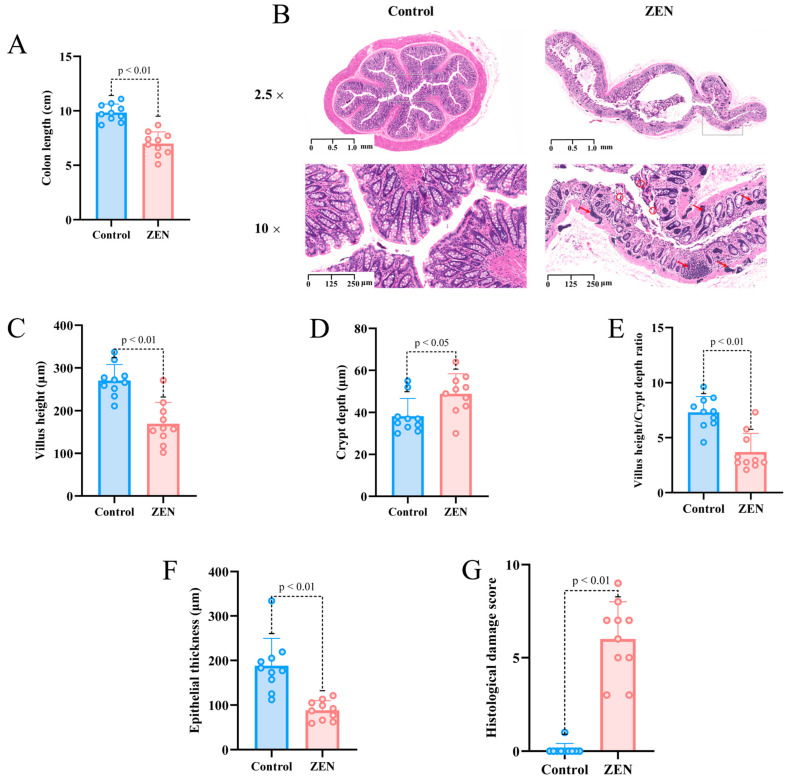
Colon length, HE section staining and microscopic observation of rats’ colon in each group (**A**) Colon length of rats in control and ZEN group; (**B**) Representative images of HE staining (2.5×) and (10×) of rat colon in control and ZEN group (The red arrow represents inflammatory infiltration, the red circle represents broken fluff); (**C**) Villus height of rats’ colon in control and ZEN group; (**D**) Crypt depth of rats’ colon in control and ZEN group; (**E**) Villus height/crypt depth ratio of rats’ colon in control and ZEN group; (**F**) Epithelial thickness of rats’ colon in control and ZEN group; (**G**) Histological damage score of rats’ colon in control and ZEN group. Data are expressed as mean ± SD. Statistical significance was determined by two-sided unpaired Student’s *t*-test. *n* = 10.

**Figure 2 toxins-15-00392-f002:**
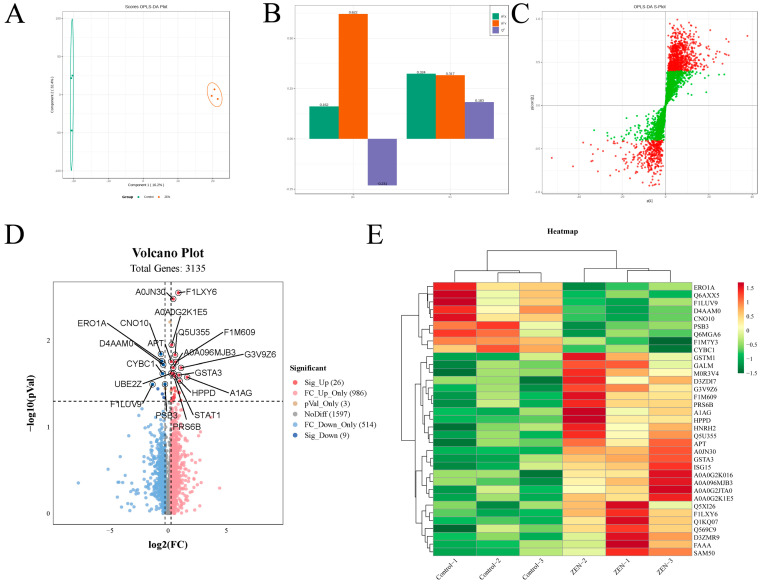
The OPLS−DA analysis, volcano plot and clustering heatmap of proteomic. (**A**) The OPLS−DA analysis of proteomic in control and ZEN group; (**B**) The OPLS−DA model validation of OPLS−DA analysis; (**C**) S−plot of OPLS−DA analysis; (**D**) The volcano plot of proteomic; (**E**) The clustering heatmap of proteomic. Data are expressed as mean ± SD. Statistical significance was determined by two-sided unpaired Student’s *t*-test. *n* = 3.

**Figure 3 toxins-15-00392-f003:**
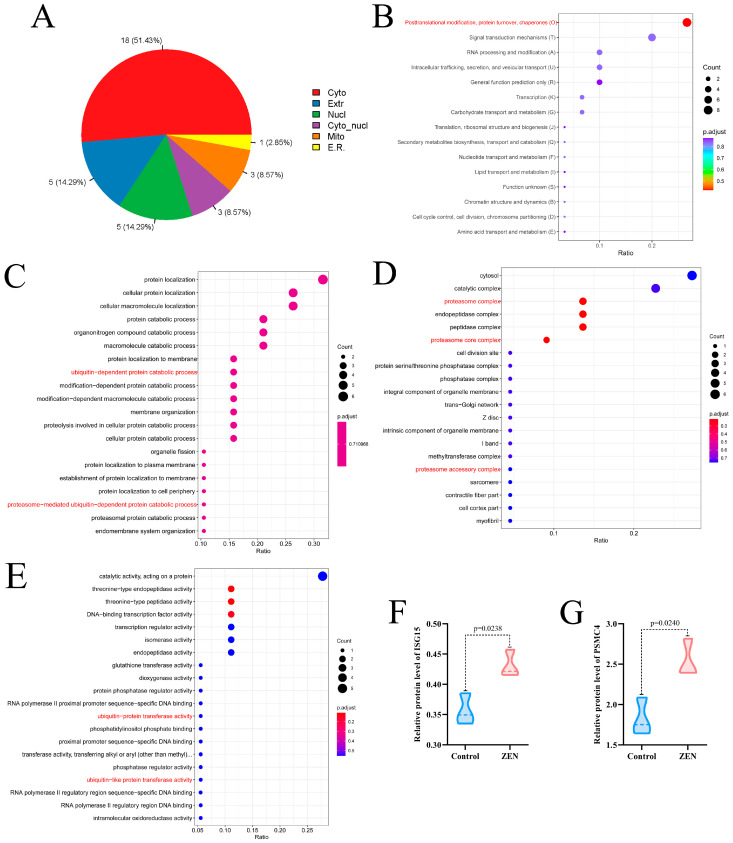
Subcellular localization, KOG enrichment analysis and GO enrichment analysis of proteomic. (**A**) Subcellular localization of proteomic; (**B**) KOG enrichment analysis of proteomic; (**C**) The biological process enrichment analysis of GO analysis; (**D**) The cell component enrichment analysis of GO analysis; (**E**) The molecular function enrichment analysis of GO analysis; (**F**) Violin plots of ISG15 between control and ZEN group; (**G**) Violin plots of PSMC4 between control and ZEN group. Data are expressed as mean ± SD. Statistical significance was determined by two-sided unpaired Student’s *t*-test. *n* = 3.

**Figure 4 toxins-15-00392-f004:**
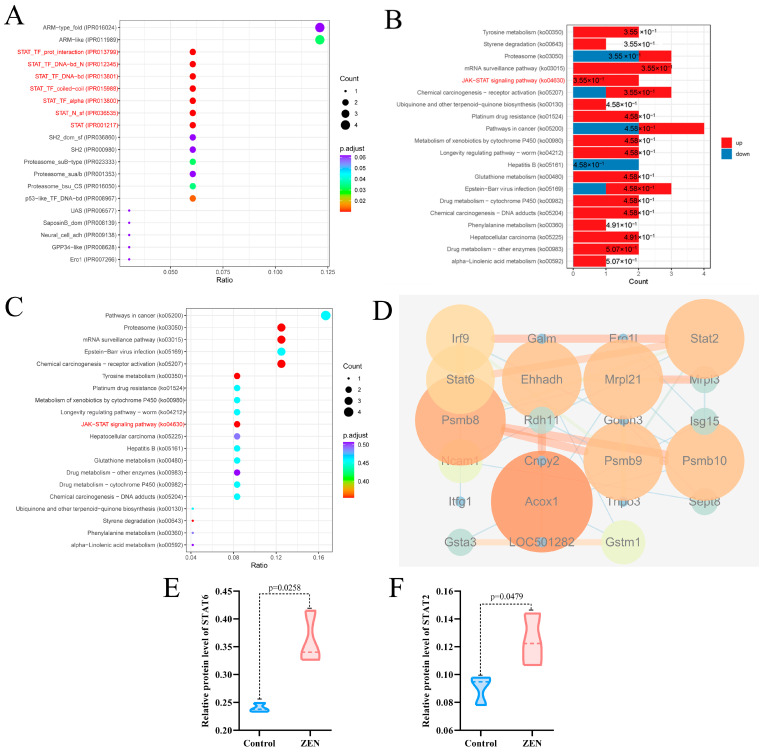
Domain enrichment analysis, KEGG enrichment analysis and PPI analysis of proteomic. (**A**) Domain enrichment analysis of proteomic; (**B**) The column chart for KEGG enrichment analysis of proteomic; (**C**) The bubble chart for KEGG enrichment analysis of proteomic; (**D**) PPI analysis of proteomic; (**E**) Violin plots of STAT6 between control and ZEN group; (**F**) Violin plots of STAT2 between control and ZEN group. Data are expressed as mean ± SD. Statistical significance was determined by two-sided unpaired Student’s *t*-test. *n* = 3.

**Figure 5 toxins-15-00392-f005:**
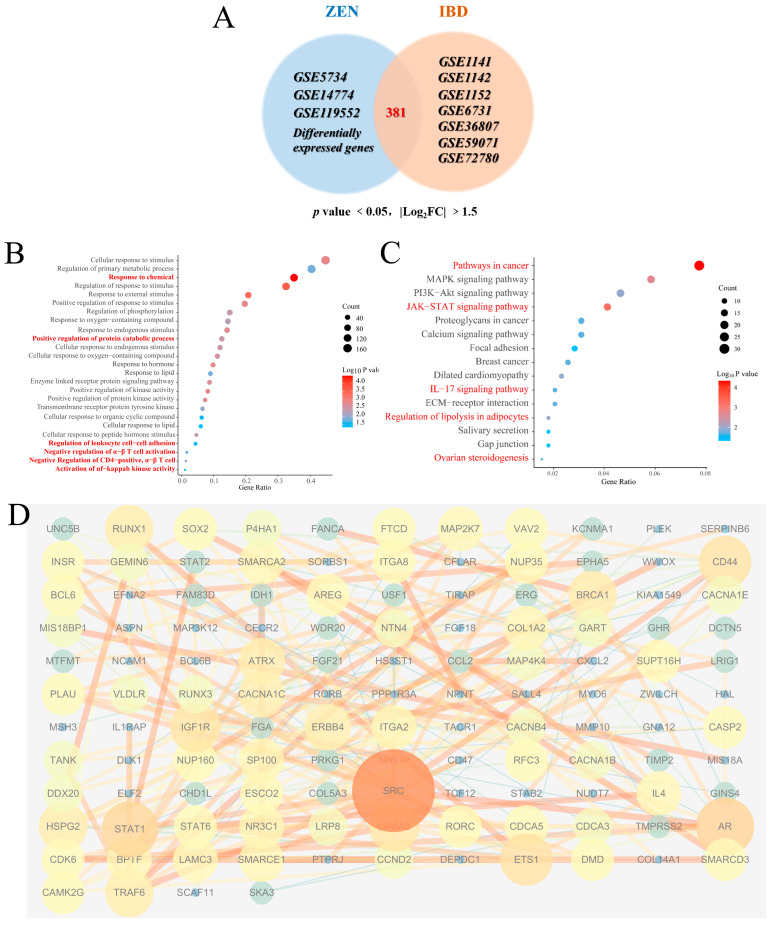
Bioinformatics analysis of ZEN exposure and IBD. (**A**) The Venn diagram of cDEGs of ZEN exposure and IBD screened from the GEO database. (**B**) The biological process of GO enrichment analysis of cDEGs of ZEN exposure and IBD screened from the GEO database. (**C**) The KEGG enrichment analysis of cDEGs of ZEN exposure and IBD screened from the GEO database. (**D**) PPI analysis of cDEGs of ZEN exposure and IBD screened from the GEO database. Data are expressed as mean ± SD. Statistical significance was determined by two-sided unpaired Student’s *t*-test.

**Figure 6 toxins-15-00392-f006:**
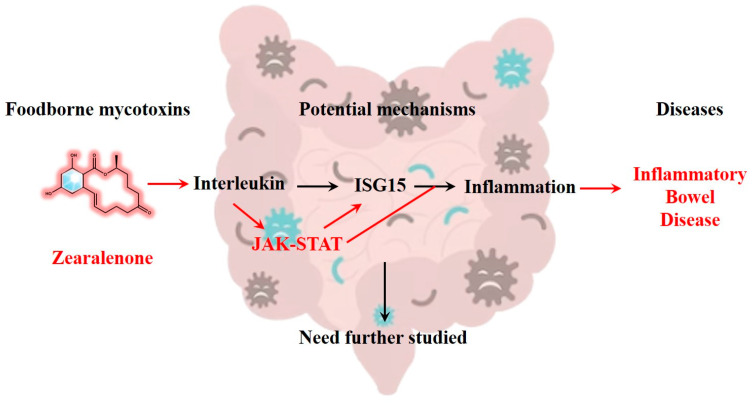
ZEN exposure may increase the risk of IBD by activating the IFN-STAT-ISG15 pathway, which needs further study.

**Table 1 toxins-15-00392-t001:** Rat colon DEPs of ZEN vs. control.

Proteins	Control-1	Control-2	Control-3	ZEN-1	ZEN-2	ZEN-3	* Log_2_FC	* *p*-Value
A1AG	0.4694	0.3347	0.791	1.1186	2.379	1.4882	1.6441	0.0264
G3V9Z6	0.4644	0.6189	0.3563	1.0716	1.1777	0.8675	1.1144	0.0206
FAAA	0.445	0.5113	0.7153	1.6298	0.7939	1.163	1.1015	0.0491
HPPD	0.301	0.3293	0.3467	0.5488	0.8308	0.5625	0.9911	0.0292
D3ZMR9	0.1233	0.146	0.1092	0.3057	0.1749	0.2517	0.9517	0.0402
GSTA3	1.475	1.5525	2.2145	3.2378	3.0798	3.6151	0.9221	0.025
GSTM1	2.166	2.2667	3.5107	5.0322	6.2371	3.7568	0.9197	0.0396
F1LXY6	0.5733	0.4813	0.5203	1.1055	0.9261	0.8807	0.8869	0.0028
F1M609	0.0766	0.1025	0.0857	0.1299	0.1656	0.127	0.6741	0.0177
GALM	0.736	1.0308	0.9585	1.5363	1.5418	1.1391	0.6299	0.0381
Q5U355	0.0961	0.1176	0.1024	0.1379	0.1793	0.1619	0.6	0.0146
Q1KQ07	0.2377	0.2491	0.2333	0.4148	0.3265	0.3403	0.587	0.0258
SAM50	0.7642	0.7959	0.9456	1.3768	1.0104	1.269	0.5451	0.0351
M0R3V4	0.3655	0.5018	0.4934	0.6545	0.7441	0.5801	0.5402	0.0452
A0A0G2JTA0	0.206	0.2746	0.2259	0.3228	0.2976	0.4062	0.539	0.0422
A0A0G2K016	0.4296	0.4338	0.3282	0.5172	0.5157	0.6476	0.4961	0.0451
PRS6B	1.6427	2.0863	1.7524	2.3933	2.8153	2.3946	0.4721	0.024
Q5XI26	0.0948	0.0781	0.0979	0.144	0.1224	0.1069	0.4632	0.0479
A0JN30	1.9871	2.0197	1.9874	2.634	2.6796	2.83	0.4421	0.0033
A0A096MJB3	0.1177	0.1115	0.0988	0.1447	0.1345	0.1658	0.4396	0.0205
HNRH2	0.6268	0.7558	0.7903	0.8522	1.0493	0.9405	0.3873	0.0456
A0A0G2K1E5	0.079	0.0864	0.0795	0.0965	0.0989	0.1089	0.3126	0.0112
D3ZDI7	0.4407	0.4408	0.3788	0.4798	0.5652	0.5167	0.3093	0.0363
APT	9.2063	10.3972	10.1311	11.2779	12.3006	12.4024	0.2751	0.0175
UB2L6	0.3352	0.3495	0.3854	0.4214	0.4154	0.4573	0.2742	0.0238
Q569C9	0.3141	0.3756	0.3559	0.4466	0.4055	0.4086	0.2698	0.0493
PSB3	3.8159	3.9469	3.4117	3.152	2.917	3.221	−0.2665	0.0319
F1M7Y3	0.9241	0.9139	0.8875	0.7159	0.7745	0.6323	−0.3606	0.0453
D4AAM0	0.3923	0.3254	0.3651	0.2561	0.2751	0.28	−0.4165	0.0189
CYBC1	0.1593	0.1766	0.165	0.1328	0.1227	0.1057	−0.4719	0.024
Q6MGA6	1.4047	1.3431	1.0569	0.9763	0.8123	0.9176	−0.4915	0.0413
ERO1A	0.3429	0.2804	0.2897	0.2111	0.1824	0.2337	−0.5415	0.0173
CNO10	0.1421	0.1117	0.1166	0.0745	0.082	0.0851	−0.6162	0.0143
Q6AXX5	0.6402	0.3931	0.4837	0.3111	0.2606	0.2541	−0.8772	0.036
F1LUV9	0.4489	0.2294	0.3003	0.0857	0.1227	0.1787	−1.3379	0.032

* FC > 1.2 or FC < 0.83, *p*-value < 0.05 were considered statistically significant.

**Table 2 toxins-15-00392-t002:** Criteria for microscopic scoring of colonic lesions.

Score	Appearance
0 None; 1 Mild; 2 Moderate; 3 Severe	Submucosal edema
0 None; 1 Localized; 2 Moderate; 3 Severe	Damage/necrosis
0 None; 1 Mild; 2 Moderate; 3 Severe	Inflammatory cell infiltration
0 None; 1 Mild; 2 Moderate; 3 Severe	Vasculitis
0 No; 1 Yes	Perforation

## Data Availability

All the data generated for this study are included in the article.

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
