# Peer review of "Zearalenone Exposure Disrupts STAT-ISG15 in Rat Colon: A Potential Linkage between Zearalenone and Inflammatory Bowel Disease"

_toxins, 2023, doi:10.3390/toxins15060392_

Round 1
Reviewer 1 Report
It is an interesting work regarding the potential correlation between zaeralenone and the STATISG15 pathway. The work is well structured however there are some points still to be clarified that require revisions:
- when you talk about the length of the colon you should specify how you carried out the measurements and what range of measurements you found (minimum-maximum) in the sample examined. There is no reference about the identified measures;
- the titles of the subparagraphs in point 3 (results) should be changed without anticipating the result which can then be discussed later;
- I would insert an image that summarizes the identified hypothesis and above all I would stress the concept that it is a hypothesis based on experimental results and that requires further studies;
- I do not read information regarding the ethics committee;
- consider to analyze the following paper: Llorens Castelló P, Sacco MA, Aquila I, Moltó Cortés JC, Juan García C. Evaluation of Zearalenones and Their Metabolites in Chicken, Pig and Lamb Liver Samples. Toxins (Basel). 2022 Nov 11;14(11):782. doi: 10.3390/toxins14110782. PMID: 36422956; PMCID: PMC9692590.
good
Reviewer 2 Report
Dear Authors,
The structure of the reviewed article is well thought out, clear, and in line with the editorial requirements of the Journal Toxins.
The introduction provides an outdated background of the topic. I think the findings of this study aren’t sufficiently described in the context of the published literature.
It is worth noting that the Authors cited in current literature in the research area.
The conclusions are supported by appropriate evidence.
I have some suggestions for Authors to improve the manuscript as follows:
· Add mean values or best results in the abstract quantitatively.
• Table 1: please, identify statistically significant differences,
· Figures: 3; 4: unreadable. Too much data, correct it. Please consider placing some of the in plots in the subsection Supplementary material.
· Please recheck thoroughly the whole article and improve its grammatical mistakes.
· All references should be cited by the same way (pages: 11-12).
· Recheck references according to the journal guidelines.
From my standpoint, this manuscript can be considered for publication in Journal – Toxins, after minor revision, given the above aspects.
Reviewer 3 Report
The paper is interesting revealing indications that zearalenone promote symptoms similar to IBD in experiments were ZEN was fed to rats. The mechanisms of colon toxicity are nicely explained, and the role of environmental pollutants and mycotoxins in development of IBD in humans are discussed.
However there are serious flaws in the paper which easily could be corrected: The font in the texts of Figures 3-5 are impossible to read. The statistics in Table 1 should be explained; are the P values presented significant should be marked . And the sentence Data are presented as average +- SD is not clear to me, where are the SD values??
Reviewer 4 Report
The study is not reported according to the ARRIVE guidelines (Percie du Sert N, Ahluwalia A, Alam S, Avey MT, Baker M, Browne WJ, et al. (2020) Reporting animal research: Explanation and elaboration for the ARRIVE guidelines 2.0. PLoS Biol 18(7): e3000411. https://doi.org/10.1371/journal.pbio.3000411.
The study is heavily biased (no blinding of subjective endpoints, no quantitative evaluation of the histopathological evaluation). Thus, the internal validity is low. The model used and the criteria used are not appropriate. Thus the external validity of the study is low.
Hence, the manuscript cannot be accepted for publication.
Round 2
Reviewer 3 Report
My only comment: the font in Fig 5 could be still enlarged if possible. This is only a suggestion
Author Response
Thanks for your nice comment. We have enlarged the font in Figure 5 to ensure the readability of the figure.
Reviewer 4 Report
The quality of the study has not improved. The scientists evaluating the results have not been blinded. Their assessment is therefore biased. Therefore the results cannot be trusted.
In addition, the authors claim in the introduction: Current studies have shown that mycotoxin exposure (including ZEN) is closely related to the occurrence and progression of IBD. This is an exaggeration of the data which are published. No study in human patients is cited. The studies cited report on animal data [Payros, D., Alassane-Kpembi, I., Laffitte, J., Wistar rats exposed (1-4 weeks) to low doses of DON (2 or 9 mg kg-1 feed) Z K Wang , Y S Yang, A T Stefka, G Sun, L H Peng (Review) : No human data. Payros, D., Ménard, S., Laffitte, J.,: Wistar rats exposed (1-4 weeks) to low doses of DON (2 or 9 mg kg-1 feed) Conclusion: highlighting oral exposure to DON as a potential risk factor in triggering IBD.] In the cited reviews the authors are extremely cautious: Maresca M, Fantini J Converging evidence based on various cellular and animal studies show that several mycotoxins induce intestinal alterations that are similar to those observed at the onset and during the progression of inflammatory bowel diseases. And Gao Y, Meng L, Liu H, Wang J, state in their review” However, potential links between mycotoxins and human chronic intestinal inflammatory diseases remain unclear”. In this context the interpretation of the data is not correctly mirroring the stae of the knowledge.
Hence, the manuscript cannot be accepted for publication.
Author Response
Thanks for your nice comment. Our experiment strictly follows the ARRIVE guidelines
According to your nice comment, we have made modifications to the parts of the article that the reviewer believe are exaggerated.